# Reduction of Osteoclastic Differentiation of Raw 264.7 Cells by EMF Exposure through TRPV4 and p-CREB Pathway

**DOI:** 10.3390/ijms24043058

**Published:** 2023-02-04

**Authors:** Myeong-Hyun Nam, Hee-Jung Park, Young-Kwon Seo

**Affiliations:** Department of Medical Biotechnology, Dongguk University, Goyang-si 10326, Republic of Korea

**Keywords:** EMF, osteoclastic differentiation, TRPV4, Ca^2+^ oscillation

## Abstract

In this study, we investigated the effect of EMF exposure on the regulation of RANKL-induced osteoclast differentiation in Raw 264.7 cells. In the EMF-exposed group, the cell volume did not increase despite RANKL treatment, and the expression levels of Caspase-3 remained much lower than those in the RANKL-treated group. TRAP and F-actin staining revealed smaller actin rings in cells exposed to EMF during RANKL-induced differentiation, indicating that EMF inhibited osteoclast differentiation. EMF-irradiated cells exhibited reduced mRNA levels of osteoclastic differentiation markers cathepsin K (CTSK), tartrate-resistant acid phosphatase (TRAP), and matrix metalloproteinase 9 (MMP-9). Furthermore, as measured by RT-qPCR and Western blot, EMF induced no changes in the levels of p-ERK and p-38; however, it reduced the levels of TRPV4 and p-CREB. Overall, our findings indicate that EMF irradiation inhibits osteoclast differentiation through the TRPV4 and p-CREB pathway.

## 1. Introduction

Osteoporosis is a skeletal disease characterized by reduced bone density. Bone health is maintained through a dynamic balance between bone formation and bone resorption [1,2]. Osteoporosis is caused by increases in bone absorption increases uncompensated by bone formation, leading to decreases in bone density [3,4]. Bone loss is associated with abnormal differentiation and proliferation of osteoclasts, leading to increased bone resorption. Accordingly, osteoclasts are considered to be potential therapeutic targets for the treatment of osteoporosis [5,6].

Calcium ion (Ca^2+^) channels are also involved in osteoclast differentiation. Several studies have suggested that the receptor activator of nuclear factors κB ligand (RANKL) induces oscillations in Ca^2+^ concentrations and activates the nuclear factor of activated T-cell c1 (NFATc1), causing osteoclast-specific gene transcription and inducing osteoclast differentiation [7,8].

Various membrane channels, including a family of transient receptor potential (TRP) ion channels in osteoclasts, control the inflow of calcium ions into the cell [9]. The family consists of TRPC and TRPV subtypes, and it has been reported that TRPV channels can regulate calcium signaling in osteoclasts [10,11]. TRPV4 is located on the membranes of various types of cells, including osteoblasts, osteoclasts, and cartilage cells. TRPV4 regulates intracellular Ca^2+^ signaling and participates in osteoclast differentiation by regulating the transcription of NFATc1 required for osteoclasts differentiation and controlling constant calcium ion inflow. Previous studies have reported that TRPV4-knockout osteoclasts exhibit reduced intracellular Ca^2+^ levels, NFATc1 expression, and osteoclast differentiation [12]. Moreover, an in vivo study has reported that TRPV4-knockout mice exhibit increased bone mass, a lower number of osteoclasts, and reduced bone resorption compared to wild-type mice [13].

The interactions between immune cells and osteoclasts have been studied in the field of osteoimmune research. Proinflammatory cytokines, such as IL-1, IL-6, and TNF-α, have considerable effects on the bone remodeling, most of which direct the system towards resorption [14]. TNF-α promotes RANKL expression in T cells [15], while IL-1 acts on osteoblasts to induce prostaglandin E2 (PGE2) synthesis, indirectly promoting osteoclast formation [16,17]. TNFα and prostaglandins play a pivotal role in osteoclast maturation. Osteoclast precursors require TNFα in the presence of small amounts of RANKL to differentiate into mature osteoclasts.

Electromagnetic-field (EMF) therapy has shown potential in the treatment of diseases of skeletal muscle as an inexpensive, safe, and non-invasive approach [18,19]. Several clinical studies have demonstrated that EMF stimulation with different frequencies and electromagnetic intensities can improve bone density, accelerate fracture healing, and decrease fracture risk [20,21,22]. Other studies have demonstrated that EMF exposure can increase bone mass in vivo, and enhance osteoblast activity and osteoblast mineralization potential in vitro, with a clear improvement in bone formation rate in osteoporotic animals [23,24,25,26]. Although many studies have been conducted reporting the effect of EMF stimulation on osteoblast formation, the regulatory mechanisms underlying the impacts of EMF exposure on osteoclast activity and function remain elusive.

In this study, we investigated the effect of EMF exposure on osteoclast differentiation induced by RANKL in Raw 264.7 macrophages and examined the molecular pathways involved in the influence of electromagnetic fields on osteoclast differentiation.

## 2. Results

### 2.1. Evaluation of Cellular Stress upon EMF Exposure

We exposed the Raw 264.7 cells to EMF (10 G, 40 Hz) and examined cell death by a lactate dehydrogenase (LDH) assay. Figure 1A shows the morphologies of untreated (negative control), RANKL-treated (positive control), and EMF-exposed RANKL-treated Raw 264.7 cells after four days. We did not observe apoptosis or necrosis in any of the three groups, including the EMF-exposed group. Figure 1B shows the cytotoxic effects of EMF exposure as measured by the LDH assay. LDH activity levels were comparable among the three groups, indicating that EMF did not induce cellular damage. Cell death was assessed by evaluating the expression levels of Caspase-3, a cell death biomarker, by quantitative PCR (qPCR). The differentiation group (positive control; treated with RANKL) exhibited increased levels of apoptosis-related Caspase-3. The EMF group, however, showed much lower Caspase-3 levels compared with the differentiation group (Figure 1C). Thus, the EMF radiation at levels tested in this study did not induce cytotoxic effects or apoptosis in Raw 264.7 cells.

### 2.2. Effect of EMF on Osteoclastic Differentiation and Cytoskeletal Organization

To explore the effect of EMF exposure on osteoclast differentiation, Raw 264.7 cells were cultured in the presence of RANKL to induce osteoclasts differentiation with or without EMF exposure. Osteoclast formation was monitored by TRAP staining (Figure 2). EMF exposure suppressed the RANKL-induced differentiation of Raw 264.7 cells into osteoclasts.

We also stained the osteoclasts with FITC-conjugated phalloidin, a cytoskeletal marker. Compared to the positive control group, EMF-exposed cells exhibited significantly reduced osteoclast spreading and smaller actin rings (Figure 3).

### 2.3. Analysis of Osteoclastic Differentiation-Related mRNA Expression

We assessed the mRNA levels of osteoclastogenic markers by RT-qPCR, to examine the effect of EMF exposure on osteoclast differentiation. Cathepsin K (CTSK) mRNA levels were significantly decreased in EMF-exposed cells compared to RANKL-treated cells (positive control; Figure 4C). Similar patterns of reduced expression were observed in TRAP, the receptor activator of nuclear factors κB (RANK) and matrix metalloproteinase 9 (MMP9) (Figure 4A,B,D). Taken together, these observations indicate that EMF inhibits RANKL-induced osteoclast differentiation.

### 2.4. Effect of EMF Exposure on Proinflammatory and Inflammatory Cytokine Levels

We measured mRNA expression levels of proinflammatory and inflammatory cytokine markers by RT-qPCR in Raw 264.7 cells and found that the mRNA levels of TNF-α, IL-1β, and IL-23 were decreased by EMF exposure compared to the positive control group (Figure 5).

### 2.5. TRPV4 and CREB Signaling Pathway Activity upon EMF Exposure

To determine the effect of EMF exposure on RANKL signaling in RANKL-treated Raw 264.7 cells, we investigated the TRPV4 levels by qPCR and p-CREB levels by Western blotting. EMF exposure reduced TRPV4 and NFATc1 mRNA levels (Figure 6A,B), and reduced the phosphorylation of CREB (Figure 6E), compared to the positive control group. However, the levels of phosphorylated ERK and p38, readouts for the MAPK signaling pathway activity in osteoclasts, did not change (Figure 6C,D).

These results indicate that EMF directly regulates RANKL-induced osteoclast differentiation by reducing the expression of NFATc1 through the reduction of TRPV4 and phosphorylated CREB levels.

## 3. Discussion

Several bisphosphates, such as alendronate and zoledronates, have been reported to reduce osteoclast formation and activity in osteoporosis [27,28]. However, the inhibition of bone resorption has been a serious health problem with the increasing number of osteoporotic patients. This is because bisphosphate can cause musculoskeletal pain, osteonecrosis of the jaw, and suppression of bone turnover [29]. So, as a new treatment for osteoporosis, several forms of physical stimuli have been studied to decrease osteoclast marker expression [30,31,32,33,34].

In our preliminary study, the osteoclastogenic-related marker expression of EMF was performed using 10 G of Raw 264.7 murine macrophages at various frequencies. During osteoclast differentiation using RANKL in Raw 264.7 cells, the mRNA expression of osteoclast-related markers (RANK, CTSK, TRAP, NFATc1, integrin b-3, and MMP-9) increases. In this study, the expression of TRAP at any frequency did not differ significantly (Appendix A). Expressions of RANK, MMP-9, and NFATc1 at a frequency of 100 Hz promoted differentiation into osteoclasts as osteoclast-related markers increased (Appendix A). However, as the expression of CTSK and integrinB-3 decreased at 40 Hz frequency, it was found that the differentiation into osteoclast was reduced (Appendix A).

Therefore, it can be inferred from previous studies that 40 Hz frequency EMF reduces osteoclastic differentiation.

In our preliminary study, the osteogenic effect of EMF was performed to various frequencies on 10 G with SaOS2 human osteoblastoma. At that study, the various frequencies of EMF-exposed did not induce apoptosis or necrosis (Appendix A). In addition, the mRNA levels of collagen Ⅲ, BMP2, PTH-R1, runx2, and osteocalcin were increased by 30 and 40 Hz of EMF exposure; however, PTH-R1 and osteocalcin were decreased by 100 Hz frequency EMF exposure compared to the control group (Appendix A). Moreover, the expression of bonesialoprotein, osteopontin, collagen I, Runx2, osteoprotegerin, and ALP increased 30 and 40 Hz frequencies EMF exposure compared with the control group by Western blotting analysis (Appendix A); especially, immunofluorescence staining of Runx2 showed that the strong fluorescence was detected in the 30, 40, and 70 Hz of the EMF-exposed group compared to the control group (Appendix A).

In addition, our previous study revealed that osteogenesis-related markers were highly expressed in the EMF-exposed Saos-2 cells. Furthermore, in vivo experiments using the rat calvarial bone defect model showed that the EMF-exposed group increased bone mass/volume, mineral density, and bone regeneration in defects compared to the control group [35].

Therefore, it can be inferred from the previous studies that 40 and 45 Hz of EMF induces osteogenesis by enhancing the activity of osteoblasts.

Recently, many investigations have been conducted on inhibiting osteoclasts activity as well as increasing osteoblast activity for the therapy of osteoporosis.

Previous studies have reported that the cell volume increases, multinucleated cells appear, and Caspase-3 is activated during the differentiation of Raw 264.7 cells into osteoclasts. In the positive control group comprised of cells cultured in differentiation media with 50 ng/mL RANKL, Raw 264.7 cells exhibited increases in cell volume and multinucleated cells appeared as expected. However, EMF exposure during the process of RANKL-induced differentiation suppressed the increases in cell volume and Caspase-3 expression levels. Moreover, Trap staining revealed that RANKL-induced osteoclast formation was reduced upon EMF exposure [36,37,38].

F-actin rings are observable in mature osteoclasts and are required for osteoclast bone resorption [39]. Previous studies have found that either the formation or the size of actin rings in osteoclasts is reduced in response to zoledronate treatment [27]. In our study, it was confirmed that the formation of actin rings was reduced in the EMF irradiation group compared to the positive control group. This result suggests that electromagnetic fields can inhibit osteoclast differentiation and bone resorption.

M-CSF and RANKL play important roles in osteoclast differentiation and maturation, and activate macrophage surface RANK receptors [40]. Raw 264.7 cells are known to be suitable for studying RANKL-induced osteoclast differentiation in vitro because they can secrete M-CSF independently to support osteoclast differentiation and exhibit high levels of RANK expression [41,42]. In several studies using bisphosphate, it has been reported that the expression levels of TRAP, MMP-9, and CTSK are decreased [27,43,44]. In this study, RANKL significantly increased the number of osteoclasts and the mRNA levels of osteoclast markers (RANK, CTSK, TRAP, and MMP-9). However, the number of osteoclasts and mRNA levels of osteoclast markers were significantly reduced upon EMF exposure (10 G, 40 Hz).

The interaction between the immune cells and osteoclasts has been studied in the field of osteoimmune research. Since proinflammatory cytokines such as TNF-α, IL-1, and IL-18 upregulate RANKL expression in T cells, they are involved in osteoclast formation by upregulating the activating effects of T cells on an osteoclast [45]. TNF-α mRNA levels increase in osteoclasts upon differentiation by RANKL treatment and can promote osteoclastic cell differentiation through autocrine signaling. In this study, we observed that the expression levels of inflammation-inducing cytokines decrease upon EMF exposure (10 G, 40 Hz). Therefore, it can be hypothesized that EMF inhibits osteoclast formation by decreasing proinflammatory cytokine expression.

Bone metabolism maintains bone mass by balancing bone resorption and bone formation [46,47,48,49]. TRPV4 directly regulates the differentiation and function of osteoclasts by mediating Ca^2+^ influx and signaling. TRPV4 mediates the sustained Ca^2+^ influx in the late stage of osteoclast differentiation, when the Ca^2+^ oscillations are no longer present [13,50]. Ca^2+^-NFATc1 signaling is an essential axis of osteoclast differentiation. TRPV4-mediated Ca^2+^ influx secures intracellular Ca^2+^ levels, ensures NFATc1-mediated gene transcription, and regulates the terminal differentiation and activity of the osteoclasts. Indeed, TRPV4^−/−^ osteoclasts exhibit reductions in intracellular Ca^2+^ concentrations, NFATc1 activity, osteoclast differentiation, and resorptive capacity [12]. Ca^2+^ oscillations activate a number of Ca^2+^/calmodulin-dependent proteins. Ca^2+^/calmodulin signaling activates CaMK-mediated cAMP reactive-element binding protein (CREB) pathway in cooperation with NFATc1, promoting osteoclast-specific gene expression. In addition, CREB contributes to the upregulation of NFATc1 expression by inducing cFOS in the AP1 complex [51].

In this study, we found that EMF exposure decreased the expression of osteoclasts markers TRAP, MMP-9, and CTSK; and, furthermore, suppressed osteoclast formation as indicated by TRAP and F-actin staining experiments. EMF exposure also reduced the levels of TRPV4 and p-CREB. These results suggest that, as the expression of TRPV4 decreases, the amount of calcium ions moving into cells decreases; and, as a result, the oscillation of calcium ions is attenuated, thereby reducing the activity of the CaMK–CREB pathway.

Our study results show that EMF exposure at an intensity of 10 G and a frequency of 40 Hz induces reductions in TRPV4 expression, intracellular calcium ion concentration, and CREB phosphorylation, resulting in the inhibition of osteoclast differentiation.

## 4. Materials and Methods

### 4.1. Cell Culture

Raw 264.7 cells were purchased from ATCC and cultured in Dulbecco’s modified Eagle’s medium (DMEM; Welgene, Daejeon, Republic of Korea) supplemented with 10% FBS and 1% penicillin–streptomycin (Welgene). Raw 264.7 is a macrophage cell line derived from male BALB/c. This line was established from a tumor induced by the Abelson murine leukemia virus.

To induce osteoclast differentiation, the cells were cultured in the presence of 50 ng/ml RANKL (PeproTech, Inc., Rocky Hill, NJ, USA) for 4 days.

### 4.2. EMF Exposure

An EMF generator was placed in a cell culture incubator (37 °C, 5% CO_2_; Figure 7) and used to apply an electromagnetic field on Raw 264.7 cells.

Raw 264.7 cells were exposed to EMF for 4 days, 3 h per day. The stimulation waveform was a sine wave with an intensity of 10 G and a frequency of 40 Hz. These magnetic field strengths and frequencies were selected based on the results of the preliminary study (Appendix A). The negative control group was cultured in a separate incubator to avoid EMF exposure.

### 4.3. Lactate Dehydrogenase Activity Assay

Cytotoxicity was evaluated using a lactate dehydrogenase assay kit (Roche, Basel, Switzerland). Briefly, 100 μL of media collected from the cell cultures on day 4 were placed in a 96-well plate and combined with 100 μL of working solution. The plate was incubated for 30 min at room temperature in the dark. The absorbance was measured at 490 nm.

### 4.4. Reverse Transcription Followed by Quantitative PCR

Total RNA was extracted using 1 mL of TRIzol reagent (Invitrogen, Waltham, MA, USA) per sample. Chloroform (200 μL; Sigma, St. Louis, MO, USA) was added, and the samples were mixed and incubated for 3 min. After centrifugation at 12,000× *g* at 4 °C for 15 min, the supernatant was transferred into a new tube and combined with 500 µL of isopropanol. The samples were incubated for 10 min, centrifuged for 10 min at 12,000× *g*, and the supernatant was discarded. Next, the pellet was washed with 1 mL of 75% ethanol and re-pelleted by centrifugation at 7500× *g* at 4 °C for 5 min. The pellet was dried at room temperature and the RNA pellet was dissolved in 20 μL of RNase-free water on ice. RNA concentration was quantified using a nanodrop spectrophotometer (Thermo Fisher Scientific, Waltham, MA, USA), and cDNA was synthesized using a reverse transcription master mix (Dynebio, Seongnam-si, Republic of Korea).

Quantitative PCR (qPCR) was performed on a StepOnePlus™ real-time PCR system (Applied Biosystems, Waltham, MA, USA) using TB Green^®^ Premix Ex Taq™ (Takara Bio, Kusatsu, Japan). Expression levels were calculated by the comparative CT (ΔΔCT) method. Primers sequences are shown in Table 1.

### 4.5. Western Blotting

To evaluate protein expression, Western blot analysis was performed after 4 days of incubation. Proteins extraction was performed as described previously [52]. Protein concentration was measured by BCA assay. For each sample, 50 μg of protein was separated at 10% SDS-PAGE 90 V for 120 min and transferred to PVDF membranes at 120 V for 90 min. The membranes were blocked in 5% skim milk dissolved in tris-buffered saline with 0.1% Tween-20 (TBST buffer) at room temperature for 1 h. After washing three times with TBST, the membrane was incubated for 1 h with primary antibodies for ERK, p-ERK, p38, p-p38, CREB, and p-CREB (cell signaling, Danvers, MA, USA). The membrane was then washed with TBST, incubated with anti-rabbit secondary antibody (2 μg) in 5% skin milk in TBS buffer for 2 h, and washed again. Enhanced chemical luminescence (ECL) solution was added on the membrane and the protein bands were visualized using a ChemiDoc XRS+ imaging system (Bio-Rad, Hercules, CA, USA).

### 4.6. Tartrate-Resistant Acid Phosphatase (TRAP) Staining

After four days of EMF irradiation, Raw 264.7 cells were washed with PBS and immobilized in 4% paraformaldehyde (PFA) for 10 min. Subsequently, the cells were stained using a TRAP staining kit (Takara, Vero Beach, FL, USA) according to the manufacturer’s instructions.

### 4.7. Immunofluorescence

Osteoclasts cultured on coverslips were fixed with 4% PFA for 10 min and permeabilized with 0.1% Triton X-100 in PBS. After washing twice with PBS, the coverslips were blocked in 0.2% BSA for 10 min and stained with FITC-conjugated phalloidin for 1 h. After washing twice with PBS, the nuclei were stained for 2 min with DAPI (0.4 μg/mL) in the dark at room temperature. Cells were washed twice with PBS, placed on a clean slide glass with Mount Fluor (BioCyc, Potsdam, Germany), and stored at 4 °C. The images were captured using a Nikon Eclipse Ti microscope(Nikon Instrument Korea, Seoul, Republic of Korea).

### 4.8. Statistical Analysis

Each experiment was performed in triplicate, and the data are presented as the mean ± standard error. Differences among the groups were analyzed by one-way analysis of variance (ANOVA) with Tukey’s post hoc test. Data that were not normally distributed were analyzed using the non-parametric Kruskal–Wallis test. Differences with *p* < 0.05 were considered significant. The symbols *, **, and *** on graphs indicate, respectively, differences with *p* < 0.05, *p* < 0.01, and *p* < 0.005.

## Figures and Tables

**Figure 1 ijms-24-03058-f001:**
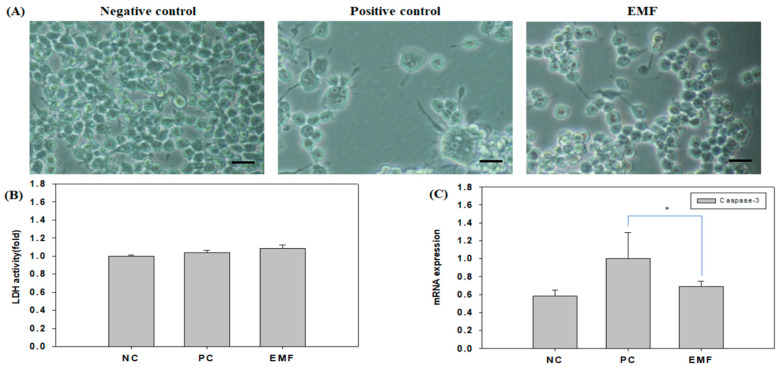
EMF exposure does not induce cellular damage. (**A**) Images of Raw 264.7 cells after EMF (10 G, 40 Hz) exposure. Scale bar, 100 μm. (**B**) Cellular stress (measured by LDH assay) and (**C**) apoptosis (measured by Caspase-3 expression) in Raw 264.7 cells after 4 days of EMF exposure. EMF, electromagnetic field; LDH, lactate dehydrogenase. Negative control (NC), cells grown in growth media (undifferentiated); positive control (PC), cells grown in differentiation media (with RANKL); EMF, cells grown in differentiation media (with RANKL) with EMF exposure. The symbol * indicate a significant difference with *p* < 0.05.

**Figure 2 ijms-24-03058-f002:**
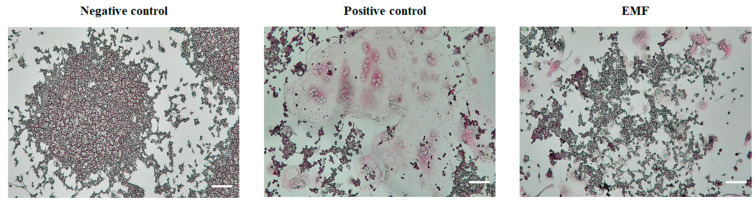
EMF exposure suppresses RANKL-mediated osteoclast differentiation. TRAP activity assay in Raw 264.7 cells differentiated into osteoclasts for 4 days in differentiation media (with RANKL) with or without EMF exposure. Images were acquired at 100× magnification. EMF, electromagnetic field; LDH, lactate dehydrogenase. Negative control, cells grown in growth media (undifferentiated); positive control, cells grown in differentiation media (with RANKL); EMF, cells grown in differentiation media (with RANKL) with EMF exposure.

**Figure 3 ijms-24-03058-f003:**
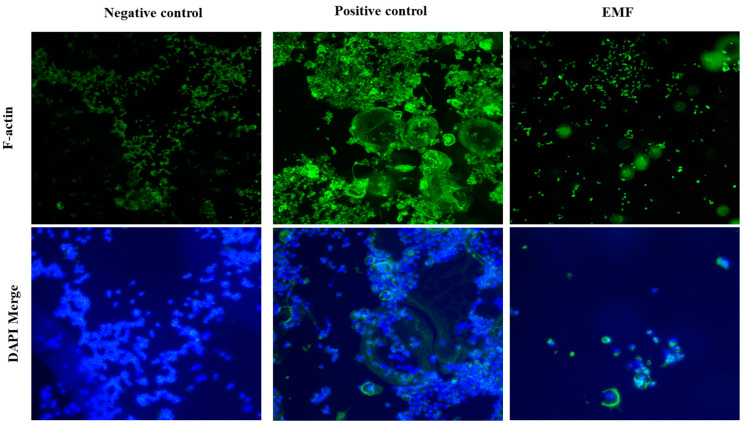
EMF-modulated actin formation. FITC-conjugated phalloidin staining in Raw 264.7 cells differentiated into osteoclasts for 4 days in differentiation media (with 50 ng/mL RANKL) with or without EMF exposure. The images were acquired at 100× magnification. Negative control, cells grown in growth media (undifferentiated); positive control, cells grown in differentiation media (with RANKL); EMF, cells grown in differentiation media (with RANKL) with EMF exposure.

**Figure 4 ijms-24-03058-f004:**
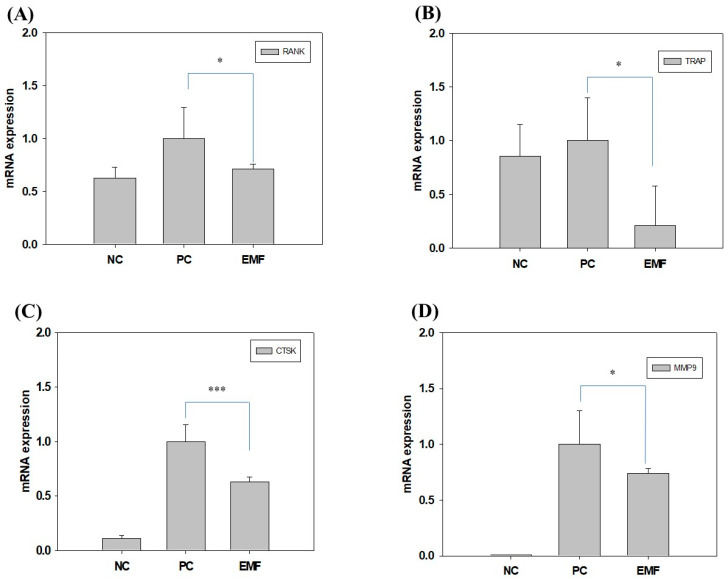
EMF regulates the expression of osteoclastogenic markers. Expression levels of (**A**) RANK, (**B**) TRAP, (**C**) CTSK, and (**D**) MMP9 were determined by RT-qPCR in Raw 264.7 cells differentiated for 4 days in differentiation media (with RANKL) with or without EMF exposure. NC, cells grown in growth media (undifferentiated); PC, cells grown in differentiation media (with RANKL); EMF, cells grown in differentiation media (with RANKL) with EMF exposure. The symbols * and *** indicate significant differences with *p* < 0.05 and *p* < 0.005, respectively.

**Figure 5 ijms-24-03058-f005:**
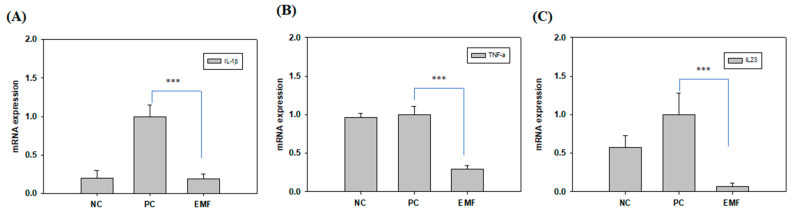
EMF-induced changes in cytokine expression. Expression levels of (**A**) IL-1β, (**B**) TNF-α, and (**C**) IL-23 were determined by RT-qPCR in Raw 264.7 cells differentiated for 4 days in differentiation media (with RANKL) with or without EMF exposure. NC, cells grown in growth media (undifferentiated); PC, cells grown in differentiation media (with RANKL); EMF, cells grown in differentiation media (with RANKL) with EMF exposure. The symbol *** indicates significant differences with *p* < 0.005.

**Figure 6 ijms-24-03058-f006:**
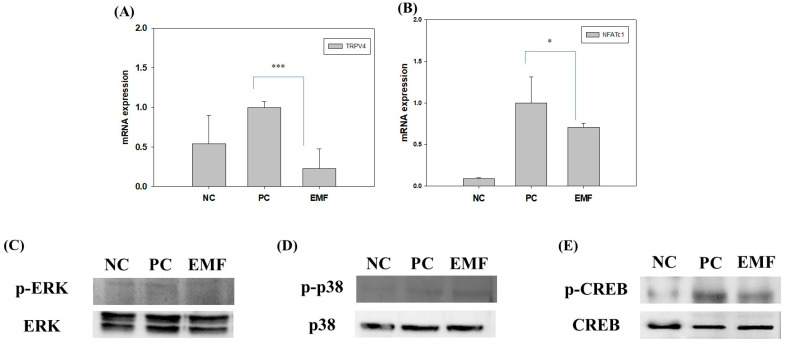
Regulation of the RANKL signaling pathway by EMF exposure. The mRNA levels of (**A**) TRPV4; and (**B**) NFATc1 and protein levels of (**C**) p-ERK, (**D**) p-p38, (**E**) p-CREB in Raw 264.7 cells differentiated for 4 days in differentiation media (with RANKL) with or without EMF exposure. NC, cells grown in growth media (undifferentiated); PC, cells grown in differentiation media (with RANKL); EMF, cells grown in differentiation media (with RANKL) with EMF exposure. The symbols * and *** indicate, respectively, significant differences with *p* < 0.05 and *p* < 0.005.

**Figure 7 ijms-24-03058-f007:**
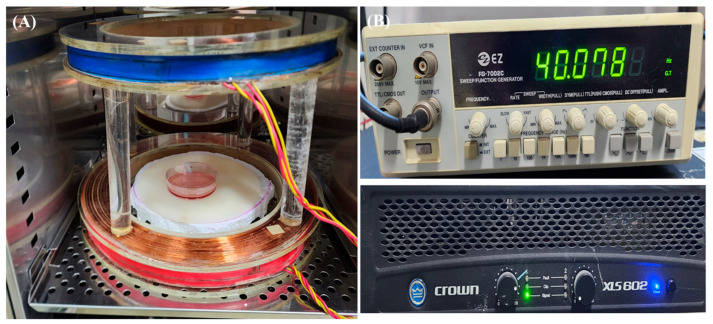
EMF stimulation device. (**A**) Helmholtz coils. (**B**) Function generator and amplifier. The coils were placed in a cell culture incubator (37 °C, 5% CO_2_). EMF, electromagnetic field.

**Table 1 ijms-24-03058-t001:** PCR primer sequences.

Gene	Forward(5′-3′)	Reverse(5′-3′)
*Mouse β-actin*	AGGCCAACCGTGAAAAGATG	TGGCGTGAGGGAGAGCATAG
*Mouse capase3*	GGCTGAAACCACCAATCGC	CTTAGCGTACCGTTCCAAGC
*Mouse NFATc1*	GACCGAGAGGCTCCGAAC	AGGGTCGAGGTGACACTAGG
*Mouse MMP-9*	AGCCGACTTTTGTGGTCTTC	AGGGTTTGCCTTCTCCGTTG
*Mouse RANK*	GAACATTGAGGACAAAGGCCC	CCACACAGGTAGGCAGTGAC
*Mouse TRAP*	GAACATTGAGGACAAAGGCCC	CCACACAGGTAGGCAGTGAC
*Mouse TNF-* *α*	CACTCACAAACCACCAAGTG	GAGTAGACAAGGTACAACCC
*Mouse IL-1* *β*	TGCCACCTTTTGACAGTGATG	TGGGTGTGCCGTCTTTCATT
*Mouse IL-23*	AAT GTG CCC CGT ATC CAG TG	GCT GCC ACT GCT GAC TAG AA
*Mouse TRPV4*	GTG ATG GTC TTT GCC CTG GT	TGA TGC CCA AGT TCT GGT TCC

RANK: receptor activator of nuclear factors κB; NFATc1: nuclear factor of activated T-cells cytoplasmic 1; MMP-9: matrix metalloproteinase-9; TRAP: tartrate-resistant acid phosphatase.

## Data Availability

The data generated and analyzed during this study are available from the corresponding author on reasonable request.

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
