# Peer review of "Reduction of Osteoclastic Differentiation of Raw 264.7 Cells by EMF Exposure through TRPV4 and p-CREB Pathway"

_ijms, 2023, doi:10.3390/ijms24043058_

Round 1
Reviewer 1 Report
The role of EMF exposure in the osteoclast differentiation remains largely unknown. I think that this study has original field. The conclusions are consistent with the evidence and arguments presented. The references are appropriate.
This manuscript was logically well written and the results were appropriately supported by the authors’ claims.
Author Response
thanks for the nice comment, Please see the attachment.

Reviewer 2 Report
The aim of the manuscript “Reduction of osteoclastic differentiation of Raw 264.7 cells by EMF exposure through TRPV4 and p-CREB pathway” present and interesting study that can be published after minor corrections.
The study is well introduced and the data presented support the efficiency of EMF to reduce osteoclast differentiation.
In the discussion the sentence “Several bisphosphates, such as alendronate and zoledronates, have been reported to reduce osteoclast formation and activity using in vitro and in vivo models of osteoporosis” is not appropriate and should be suppressed.
Moreover the authors should not forget that a too important reduction of osteoclast activity reduce osteoblast activity. Some medications that too importantly reduce osteoclast activity induce facture because of a too important reduction of bone metabolism.
Even if as reported by the authors “several clinical studies have demonstrated that EMF stimulation with different frequencies and electromagnetic intensities can improve bone density, accelerate fracture healing, and decrease fracture risk” the efficiency of EMF in an in vivo model of osteoporosis should be evaluated to support its efficiency to increase bone mineral density.
A sentence concerning the perspective open by this study should added.
Author Response
Thaks for the nice comment. Please see the attachment.
